# Regularized Maximum Diversification Investment Strategy †

**N'Golo Koné**

Department of Economics, Queen's University, 94 University Avenue Kingston, Kingston, ON K7L 3N6, Canada; ngk1@queensu.ca; Tel.: +1-514-652-1005
† I am greatly indebted to Marine Carrasco for her invaluable guidance. I thank Georges Dionne, Tony S. Wirjanto, Jade Wei, and two anonymous referees for their helpful comments.

**Abstract:** The maximum diversification has been shown in the literature to depend on the vector of asset volatilities and the inverse of the covariance matrix of the asset return. In practice, these two quantities need to be replaced by their sample statistics. The estimation error associated with the use of these sample statistics may be amplified due to (near) singularity of the covariance matrix, in financial markets with many assets. This, in turn, may lead to the selection of portfolios that are far from the optimal regarding standard portfolio performance measures of the financial market. To address this problem, we investigate three regularization techniques, including the ridge, the spectral cut-off, and the Landweber–Fridman approaches in order to stabilize the inverse of the covariance matrix. These regularization schemes involve a tuning parameter that needs to be chosen. In light of this fact, we propose a data-driven method for selecting the tuning parameter. We show that the selected portfolio by regularization is asymptotically efficient with respect to the diversification ratio. In empirical and Monte Carlo experiments, the resulting regularized rules are compared to several strategies, such as the most diversified portfolio, the target portfolio, the global minimum variance portfolio, and the naive 1/N strategy in terms of in-sample and out-of-sample Sharpe ratio performance, and it is shown that our method yields significant Sharpe ratio improvements.

**Keywords:** portfolio selection; maximum diversification; regularization

**JEL Classification:** G11; C16; C52

## 1. Introduction

Since the seminal work of Markowitz (1952), which offers essential basis to portfolio selection, diversification issues have been in the center of many problems in the financial market. According to Markowitz's portfolio theory, a portfolio is diversified if its variance could not be reduced any further at the same level of the expected return. The fundamental objective of this diversification is to construct a portfolio with various assets that earns the highest return for the least volatility that may be a good alternative to the market cap-weighted portfolios. In fact, there is evidence that market portfolios are not as efficient as assumed by Sharpe (1964) in his Capital Asset Price Model (CAPM). The CAPM model as introduced by Sharpe (1964) implies that the tangency portfolio is the only efficient one and should produce the greatest returns relative to risk. Nonetheless, several empirical studies have shown that investing in the minimum variance portfolio yields better out-of-sample results than does an investment in the tangency portfolio (see Haugen and Baker 1991; Choueifaty et al. 2013; Lohre et al. 2014).

Even if these surprising results seem to be due to the high estimation risk associated with the expected returns (according to Kempf and Memmel (2006)), the efficiency of the market capitalization-weighted index has been questioned motivating numerous investment alternatives (see Arnott et al. (2005); Clarke et al. (2006); Maillard et al. (2010)). Subsequently, Choueifaty (2011) introduced the concept of maximum diversification, via a formal definition of portfolio diversification: the diversification ratio (DR) and claimed

that portfolios with maximal DRs were maximally diversified and provided an efficient alternative to market cap-weighted portfolios.

This optimal maximum diversification portfolio is shown to be a function of the inverse of the covariance matrix of asset returns (see Theron and Van Vuuren 2018), which is unknown and needs to be estimated. Solving for the maximum diversification portfolio leads to estimate the covariance matrix of returns and take its inverse. This results in estimation error, amplified due to (near) singularity of the covariance matrix, in financial markets with many assets. This, in turn, may lead to the selection of portfolios that are far from the optimal regarding standard portfolio performance measures of the financial market. Therefore, Choueifaty et al. (2013) propose the most diversified portfolio (MDP) by imposing a non-negative constraint on the maximum diversification problem[1]. However, this ad hoc constraint suggests that the MDP is unlikely to represent the final word of diversification. Without the ability to short securities it may be impossible to unlock the full range of uncorrelated risk sources present in the market (see Maguire et al. 2014). Therefore, this paper proposes a more general method to control for estimation error in the covariance matrix of asset returns without restricting the ability to short sell in the financial market. This method is fundamentally based on different ways to stabilize the inverse of the covariance matrix particularly useful when the number of assets in the financial market increases considerably compared with the estimation window. More precisely, as in Carrasco (2012) and Carrasco and Tchuente (2015) we investigate three regularization techniques including the spectral cut-off, the Tikhonov, and Landweber–Fridman approaches in order to stabilize the inverse of the covariance matrix. This procedure has been used by Carrasco et al. (2019) to stabilize the inverse of the covariance matrix in the mean-variance portfolio.

These regularization schemes involve a tuning parameter that needs to be chosen. Therefore, we propose a data-driven method for selecting the tuning parameter that minimizes the distance between the inverse of the estimated covariance matrix and the inverse of the population covariance matrix.

We show, under appropriate regularity conditions, that the selected strategy by regularization is asymptotically efficient with respect to the diversification ratio for a wide choice of the tuning parameter. Meaning that, even if the optimal diversified portfolio is unknown, there exists a feasible portfolio obtained by regularization capable of reaching similar level of performance in terms of the diversification ratio.

To evaluate the performance of our procedures, we implement a simulation exercise based on a three-factor model calibrated on real data from the US financial market. We obtain by simulation that our procedure significantly improve the performance of the proposed strategy with respect to the Sharpe ratio. Moreover, the regularized rules are compared to several strategies such as the most diversified portfolio, the target portfolio, the global minimum variance portfolio, and the naive 1/N strategy in terms of in-sample and out-of-sample Sharpe ratio, and it is shown that our method yields significant Sharpe ratio improvements. To confirm our simulations, we do an empirical analysis using Kenneth R. French's 30-industry portfolios, 100 portfolios formed on size and book-to-market, and a subset of the S&P500 index constituents. The empirical results show that by stabilizing the inverse of the covariance matrix in the maximum diversification portfolio, we considerably improve the performance of the selected strategy in terms of maximizing the Sharpe ratio.

The main finding of this paper is that by stabilizing the inverse of the covariance matrix in the maximum diversification portfolio, we considerably improve the performance of the selected portfolio with respect to several statistics in the financial market including the diversification ratio, the Sharpe ratio, and the rebalancing costs (turnover) as shown by extensive simulations and empirical study. Therefore, our methods are highly recommended for investors in the sense that these procedures help them to select very effective strategies with lower rebalancing cost.

---

[1] The objective is to reduce the effect of estimation error on the performance of selected maximum diversification portfolio.

The rest of the paper is organized as follows. Section 2 presents the economy. The regularized portfolio is presented in Section 3. Section 4 gives some asymptotic properties of the selected strategy and proposes a data-driven method to select the tuning parameter. Section 5 presents some simulation results and an empirical study. Section 6 concludes the paper.

## 2. The Model

We consider a simple economy with $N$ risky assets with random returns vector $R_{t+1}$ and a risk-free asset. Let us denote $R_f$ the gross return on this risk-free asset. Empirically with monthly data, $R_f$ is calibrated to be the mean of the one-month Treasury-Bill (T-B) rate observed in the data. The number of risky assets in our economy $N$ is assumed to be large for diversification issue.

We assume that the excess returns $r_{t+1} = R_{t+1} - R_f 1_N$ are independent and identically distributed with the mean and the covariance matrix given by $\mu$ and $\Sigma = \{\sigma_{i,j}\}_{i,j \in N}$, respectively. Let us denote by $\omega = (\omega_1, ..., \omega_N)'$ the vector of portfolio weights that represents the amount of the capital to be invested in the risky assets and the remain $1 - \omega' 1_N$ is allocated to the risk-free asset. Short-selling is allowed in the financial market, i.e., some of the weights $\omega_i$ could be negative. Let us denote $\sigma = (\sigma_{1,1}, ..., \sigma_{N,N})'$ the vector of asset volatilities.

According to Choueifaty (2011), the diversification ratio (DR) of any portfolio $\omega$ is given by

$$DR(\omega) = \frac{\omega' \sigma}{\sqrt{\omega' \Sigma \omega}} \tag{1}$$

which is the ratio of weighted average of volatilities divided by the portfolio volatility.

Using the relation in Equation (1), the maximum diversification portfolio is obtained by solving the following optimization problem

$$\max_{\omega} DR(\omega). \tag{2}$$

As the DR is invariant by scalar multiplication (for instance see Choueifaty et al. (2013)), solving the problem in Equation (2) is equivalent of solving the following new problem according to Theron and Van Vuuren (2018)

$$\min_{\omega' \sigma = 1} \frac{1}{2} \omega' \Sigma \omega. \tag{3}$$

This new optimization problem is very close to the global minimum variance portfolio. The only difference is that the constraint $\omega' 1 = 1$ in the global minimum variance problem is replaced by $\omega' \sigma = 1$. The solution of this new optimization problem is given by

$$\omega = \frac{\Sigma^{-1} \sigma}{\sigma' \Sigma^{-1} \sigma} = \left(\sigma' \Sigma^{-1} \sigma\right)^{-1} \left(\Sigma^{-1} \sigma\right). \tag{4}$$

The solution in (4) is unknown because it depends on the covariance matrix of asset returns and the vector of volatilities that are unknown and need to be estimated from available data set. We need, in particular, to estimate the covariance matrix and take its inverse. The sample covariance may not be appropriate because it may be nearly singular, and sometimes not invertible. The issue of ill-conditioned covariance matrix must be addressed because inverting such matrix increases dramatically the estimation error and then makes the maximum diversification portfolio unreliable. Many techniques have been

proposed in the literature to stabilize the inverse of the covariance matrix in the solution in (4). According to Carrasco et al. (2007), an interesting way to stabilize the inverse of the covariance matrix consists of dampening the explosive effect of the inversion of the singular values of $\hat{\Sigma}$. It consists in replacing the sequence $\{1/\lambda_j\}$ of explosive inverse singular values by a sequence $\{q(\alpha, \lambda_j)/\lambda_j\}$, where the damping function $q(\alpha, \lambda)$ is chosen such that

1. $q(\alpha, \lambda)/\lambda$ remains bounded when $\lambda \to 0$
2. for any $\lambda$, $\lim_{\alpha \to 0} q(\alpha, \lambda) = 1$

where $\alpha$ is the regularization parameter. The damping function is specific to each regularization.

In this paper, we propose a consistent way to estimate the solution in (4) using three regularization schemes based on three different ways of inverting the covariance matrix of asset returns. These regularization techniques are the spectral cut-off, the Tikhonov, and the Landweber–Fridman. The spectral cut-off regularization scheme is based on principal components whereas the Tikhonov's one is based on Ridge regression (also called Bayesian shrinkage) and the last one is an iterative method.

## 3. The Regularized Portfolio

The regularization methods used in this paper are drawn from the literature on inverse problems (see Kress (1999)). They are designed to stabilize the inverse of Hilbert–Schmidt operators (operators for which the eigenvalues are square summable). These regularization techniques will be applied to the sample covariance matrix of asset returns to stabilize its inverse in the selected portfolio.

Let $\hat{\lambda}_1^2 \geq \hat{\lambda}_2^2 \geq ... \geq \hat{\lambda}_N^2 \geq 0$ be the eigenvalues of the sample covariance matrix $\hat{\Sigma}$. By spectral decomposition, we have that $\hat{\Sigma} = PDP'$ with $PP' = I_N$, where $P$ is the matrix of eigenvectors and $D$ the diagonal matrix with eigenvalues $\hat{\lambda}_j^2$ on the diagonal. Furthermore, let $\hat{\Sigma}^\alpha$ be the regularized inverse of $\hat{\Sigma}$.

$$\hat{\Sigma}^\alpha = PD^\alpha P'$$

where $D^\alpha$ is the diagonal matrix with elements $q(\alpha, \hat{\lambda}_j^2)/\hat{\lambda}_j^2$. The positive parameter $\alpha$ is the regularization parameter, a kind of smoothing parameter which is unknown and needs to be selected. $q(\alpha, \hat{\lambda}_j^2)$ is the damping function that depends on the regularization scheme used.

### 3.1. Tikhonov Regularization (TH)

This regularization scheme is close to the well known ridge regression used in presence of multicollinearity to improve properties of OLS estimators. In Tikhonov's regularization scheme, the real function $q(\alpha, \hat{\lambda}_j^2)$ is given by

$$q(\alpha, \hat{\lambda}_j^2) = \frac{\hat{\lambda}_j^2}{\hat{\lambda}_j^2 + \alpha}$$

### 3.2. The Spectral Cut-Off (SC)

It consists in selecting the eigenvectors associated with the eigenvalues greater than some threshold.

$$q(\alpha, \hat{\lambda}_j^2) = I\left\{\hat{\lambda}_j^2 \geq \alpha\right\}$$

The explosive influence of the factor $1/\hat{\lambda}_j^2$ is filtered out by imposing $q(\alpha, \hat{\lambda}_j^2) = 0$ for small $\hat{\lambda}_j^2$, that is, $\hat{\lambda}_j^2 < \alpha$. $\alpha$ is a positive regularization parameter such that no bias is introduced

when $\hat{\lambda}_j^2$ exceeds the threshold $\alpha$. Another version of this regularization scheme is the Principal Components (PC) which consists in using a certain number of eigenvectors to compute the inverse of the operator. The PC and the SC are perfectly equivalent, only the definition of the regularization term $\alpha$ differs. In the PC, $\alpha$ is the number of principal components. In practice, both methods will give the same estimator.

### 3.3. Landweber–Fridman Regularization (LF)

In this regularization scheme, $\hat{\Sigma}^{\alpha}$ is computed by an iterative procedure with the formula

$$\begin{cases} \hat{\Sigma}_l^{\alpha} = (I_N - c\hat{\Sigma}^{\alpha})\hat{\Sigma}_{l-1} + c\hat{\Sigma} & \text{for } l = 1, 2, ... 1/\alpha - 1 \\ \hat{\Sigma}_0^{\alpha} = c\hat{\Sigma} \end{cases}$$

The constant $c$ must satisfy $0 < c < 1/\hat{\lambda}_1^2$. Alternatively, we can compute this regularized inverse with

$$q(\alpha, \hat{\lambda}_j^2) = 1 - \left(1 - c\hat{\lambda}_j^2\right)^{\frac{1}{\alpha}}$$

The basic idea behind this procedure is similar to the spectral cut-off method but with a smooth bias function.

See Carrasco et al. (2007) for more details about these regularization techniques. The regularized diversified portfolio for a given regularization scheme is

$$\hat{\omega}_{\alpha} = \frac{\hat{\Sigma}^{\alpha}\hat{\sigma}}{\hat{\sigma}'\hat{\Sigma}^{\alpha}\hat{\sigma}} = \left(\hat{\sigma}'\hat{\Sigma}^{\alpha}\hat{\sigma}\right)^{-1}\hat{\Sigma}^{\alpha}\hat{\sigma}. \tag{5}$$

This regularized portfolio depends on an unknown tuning parameter that needs to be selected through a data-driven method.

## 4. Asymptotic Properties of the Selected Portfolio

In this section, we will look at the efficiency of the regularized portfolio with respect to the diversification ratio. We will also propose a data-driven method to select the tuning parameter.

### 4.1. Efficiency of the Regularized Diversified Portfolio

To obtain the efficiency of the selected portfolio, we need to impose some regularity conditions, in particular we will need the following assumption.

**Assumption A:** $\frac{\Sigma}{N}$ is a trace class operator.

A a trace class operator $K$ is a compact operator with a finite trace, i.e., $Tr(K) = O(1)$. This assumption is more realistic than assuming that $\Sigma$ is a Hilbert–Schmidt operator. Moreover, Carrasco et al. (2019) show that Assumption A holds when the returns are generated from a standard factor model.

Under Assumption A, the following proposition presents information about the asymptotic property of the diversification ratio associated with the selected portfolio.

**Proposition 1.** *Under Assumption A we have that*

$$DR(\hat{\omega}_{\alpha}) \to_p DR(\omega_t), \tag{6}$$

*if* $\frac{N}{\alpha\sqrt{T}} \to 0$ *as T goes to infinity.*

**Proof.** In Appendix A. □

**Comment on Proposition 1.** The regularity condition behind proposition 1 implies several things: First, $\alpha\sqrt{T} \to +\infty$ implies that the estimation window should go to infinity faster than the optimal tuning parameter goes to zero. Second, $\frac{N}{\alpha\sqrt{T}} \to 0$ implies that $\alpha\sqrt{T}$ should go to infinity faster than the number of assets in the financial market. Therefore, the number of assets should be limited asymptotically compared with the estimation window. As the regularization parameter $\alpha$ is in $(0,1)$, $\frac{N}{\alpha\sqrt{T}} \to 0$ is implied by the following condition $\frac{N}{\sqrt{T}} \to 0$. However, the regularity condition $\frac{N}{\sqrt{T}} \to 0$ seems to be more restrictive than assuming that $\frac{N}{T} \to Constant$. One way to avoid this regularity condition will be to assume that the covariance matrix of assets returns is a trace class operator or to assume that this covariance matrix is a Hilbert–Schmidt operator. These assumptions seem to be more restrictive than assuming that $\frac{N}{\sqrt{T}} \to 0$, which seems to be close to the reality asymptotically. Moreover, $\frac{N}{\sqrt{T}} \to 0$ is only an asymptotic assumption and we do not need to have $\frac{N}{\sqrt{T}}$ close to zero in practice to obtain good performance with the regularized portfolio. Particularly, in finite sample, $N$ could be larger than $T$ or too close to $T$. Proposition 1 shows that the regularized diversified portfolio is asymptotically efficient in terms of the diversification ratio for a wide choice of the tuning parameter. Meaning that, even if the optimal diversified portfolio in Equation (4) is unknown, there exists a feasible portfolio obtained by regularization capable of reaching similar level of performance in terms of the diversification ratio.

*4.2. Data-Driven Method for Selecting the Tuning Parameter*

We show in the previous sections that the selected portfolio depends on a certain smoothing parameter $\alpha \in (0,1)$. We have derived the efficiency of the selected portfolio assuming that this tuning parameter is given. However, in practice, the regularization parameter is unknown and needs to be selected. Therefore, we propose a data-driven selection procedure to obtain an approximation of this parameter.

Our objective here is to select the tuning parameter which minimizes the distance between the inverse of the estimated covariance matrix and the inverse of the true covariance matrix. According to Ledoit and Wolf (2003), most of the existing shrinkage estimators from finite-sample statistical decision theory as well as in Frost and Savarino (1986) break down when $N \geq T$ because their loss functions involve the inverse of the sample covariance matrix which is a singular matrix in this situation. Therefore, to avoid this problem, they propose a loss function that does not depend on this inverse. This loss function is a quadratic measure of distance between the true and the estimated covariance matrices based on the Frobenius norm. Unlike in Ledoit and Wolf (2003), we will use a loss function that depends on the inverse of the covariance matrix under the assumption that the true covariance matrix is invertible. One important thing to notice here is that the regularized covariance matrix is always invertible even if $N \geq T$ meaning that our loss function exists for $N \geq T$. In fact, we know that the optimal diversified portfolio as given by Equation (4) depends on the inverse of the covariance matrix of assets returns. Moreover, because our objective is to stabilize the inverse of this covariance matrix in the estimated portfolio by regularization, we propose to use a loss function that minimizes a quadratic distance between the regularized inverse and the theoretical covariance matrix.

The loss function we consider here is given by

$$\mu'\left[\left(\hat{\Sigma}^{\alpha} - \Sigma^{-1}\right)'\Sigma\left(\hat{\Sigma}^{\alpha} - \Sigma^{-1}\right)\right]\mu \tag{7}$$

where $\mu$ is the expected excess return. The choice of this specific quadratic distance is useful to obtain a criterion that can easily be approximated by generalized cross validation approach.

Therefore, the objective is to select the tuning parameter that minimizes

$$E\left\{\mu'\left[\left(\hat{\Sigma}^{\alpha}-\Sigma^{-1}\right)'\Sigma\left(\hat{\Sigma}^{\alpha}-\Sigma^{-1}\right)\right]\mu\right\}. \tag{8}$$

It implies that

$$\hat{\alpha}=arg\min_{\alpha\in H_T}E\left\{\mu'\left[\left(\hat{\Sigma}^{\alpha}-\Sigma^{-1}\right)'\Sigma\left(\hat{\Sigma}^{\alpha}-\Sigma^{-1}\right)\right]\mu\right\} \tag{9}$$

To obtain a better approximation of the tuning parameter based on a generalized cross-validation criterion, we need additional assumptions. Therefore, let us start with some useful notations.

We denote by $\Omega=E\left(r_t r_t'\right)=E\left(X'X\right)/T$ and $\beta=\Omega^{-1}\mu=E(X'X)^{-1}E(X'1_T)$ where $r_t, t=1,\cdots,T$ are the observations of the excess returns and $X$ the $T\times N$ matrix with $t$th row given by $r_t'$.

**Assumption B**

For some $\nu>0$, we have that

$$\sum_{j=1}^{N}\frac{<\beta,\phi_j>^2}{\eta_j^{2\nu}}<\infty$$

where $\phi_j$ and $\eta_j^2$ denote the eigenvectors and eigenvalues of $\frac{\Omega}{N}$.

The regularity condition in Assumption B can be found in Carrasco et al. (2007) and Carrasco (2012). Moreover, Carrasco et al. (2019) show that Assumption B hold if the returns are generated by a factor model. Assumption B is used combined with Assumption A to derive the rate of convergence of the mean squared error in the OLS estimator of $\beta$. These two assumptions imply in particular that $\|\beta\|^2<+\infty$ such that we have the following relations,

$$\|\beta-\beta_\alpha\|^2=\begin{cases}O(\alpha^\nu)&\text{for }SC,LF\\O\left(\alpha^{min(\nu,2)}\right)&\text{for }T\end{cases}$$

$\beta_\alpha$ is the regularized version of $\beta$.

The following result gives us a very nice equivalent of the objective function. We can easily apply a cross-validation approximation procedure on this expression of the objective function.

**Proposition 2.** *Under Assumptions A and B we have that*

$$E\left\{\mu'\left[\left(\hat{\Sigma}^{\alpha}-\Sigma^{-1}\right)'\Sigma\left(\hat{\Sigma}^{\alpha}-\Sigma^{-1}\right)\right]\mu\right\}\sim E\left\{\left(\hat{\Sigma}^{\alpha}\hat{\mu}-\Sigma^{-1}\mu\right)'\Sigma\left(\hat{\Sigma}^{\alpha}\hat{\mu}-\Sigma^{-1}\mu\right)\right\}$$

*if $\frac{1}{\alpha^2 T}\to 0$ and $\sqrt{N}\alpha^{\min(\frac{\nu}{2},1)}\to 0$ as $T$ goes to infinity.*

**Proof.** In Appendix B. □

We obtain the following corollary from this proposition.

**Corollary 1.** *Under Assumptions A and B we have that*

$$E\left\{\mu'\left[\left(\hat{\Sigma}^{\alpha}-\Sigma^{-1}\right)'\Sigma\left(\hat{\Sigma}^{\alpha}-\Sigma^{-1}\right)\right]\mu\right\}\sim\frac{1}{T}E\|X(\hat{\beta}_\alpha-\beta)\|^2+\frac{(\mu'(\beta_\alpha-\beta))^2}{(1-\mu'\beta)}$$

*if $\frac{1}{\alpha^2 T}\to 0$ and $\sqrt{N}\alpha^{\min(\frac{\nu}{2},1)}\to 0$ as $T$ goes to infinity.*

The result in Corollary 1 is obtained by using Proposition 2 combined with Proposition 1 in Carrasco et al. (2019).

From Corollary 1, it follows that minimizing $E\left\{\mu'\left[\left(\hat{\Sigma}^\alpha - \Sigma^{-1}\right)'\Sigma\left(\hat{\Sigma}^\alpha - \Sigma^{-1}\right)\right]\mu\right\}$ is equivalent to minimizing

$$\frac{1}{T}E\left\|X(\hat{\beta}_\alpha - \beta)\right\|^2 \tag{10}$$

$$+\frac{(\mu'(\beta_\alpha - \beta))^2}{(1 - \mu'\beta)}. \tag{11}$$

Terms (10) and (11) depend on the unknown $\beta$, and therefore need to be approximated. The approximation of these two quantities is borrowed from Carrasco et al. (2019). More precisely, the rescaled MSE

$$\frac{1}{T}E\left[\left\|X\left(\hat{\beta}_\alpha - \beta\right)\right\|^2\right]$$

can be approximated by generalized cross-validation criterion:

$$GCV(\alpha) = \frac{1}{T}\frac{\|(I_T - M_T(\alpha))1_T\|^2}{(1 - tr(M_T(\alpha))/T)^2}.$$

Using the fact that

$$\hat{\mu}'(\beta_\alpha - \beta) = \frac{1_T'}{T}(M_T(\alpha) - I_T)X\beta,$$

(11) can be estimated by plug-in:

$$\frac{\left(1_T'(M_T(\alpha) - I_T)X\hat{\beta}_{\tilde{\alpha}}\right)^2}{T^2\left(1 - \hat{\mu}'\hat{\beta}_{\tilde{\alpha}}\right)} \tag{12}$$

where $\hat{\beta}_{\tilde{\alpha}}$ is an estimator of $\beta$ obtained for some consistent $\tilde{\alpha}$ ($\tilde{\alpha}$ can be obtained by minimizing $GCV(\alpha)$).

The optimal value of $\tau$ is defined as

$$\hat{\alpha} = \arg\min_{\tau \in H_T}\left\{GCV(\alpha) + \frac{\left(1_T'(M_T(\alpha) - I_T)X\hat{\beta}_{\tilde{\alpha}}\right)^2}{T^2\left(1 - \hat{\mu}'\hat{\beta}_{\tilde{\alpha}}\right)}\right\}$$

where $H_T = \{1, 2, ..., T\}$ for spectral cut-off and Landweber–Fridman and $H_T = (0, 1)$ for Ridge.

## 5. Simulations and Empirical Study

We start this section by a simulation exercise to set up the performance of our procedure and compare our result to the existing methods. In particular, we compare our method to the most diversified portfolio proposed by Choueifaty and Coignard (2008). More precisely, we focus on how our procedure performs in terms of the Sharpe ratio and the diversification ratio. To end this section, we analyze the out-of-sample performance of the selected portfolio.

### 5.1. Data

In our simulations and empirical analysis, various forms of monthly data will be used from July 1980 to June 2016. The one-month Treasury-Bill (T-Bill) rate is used as a proxy for the risk-free rate, and $R_f$ is calibrated to be the mean of the one-month Treasury-Bill rate observed in the data. We use monthly returns of Fama–French three factors and of 30 industry portfolios from the Kenneth R. French data library in order to calibrate unknown

parameters of the simulation model. In the empirical study, we also use monthly data for the 100 portfolios formed on size and book-to-market from the Kenneth R. French data Library and the CRSP monthly data for the S&P500 index constituents.

*5.2. Simulation*

We implement a simple simulation exercise to assess the performance of our procedure and compare it with the existing procedures. Let us consider for this purpose a simple economy with $N \in \{10, 20, 40, 60, 80, 90, 100\}$ risky assets. We use several values of $N$ to see how the size of the financial market (defined by the number of assets in the economy) could affect the performance of the selected strategy. Let $T$ be the sample size used to estimate the unknown parameters in the investment process. Following Chen and Yuan (2016) and Carrasco et al. (2019), we simulate the excess returns at each simulation step from the following three-factor model for $i = 1, ..., N$ and $t = 1, ..., T$

$$r_{it} = b_{i1}f_{1t} + b_{i2}f_{2t} + b_{i3}f_{3t} + \epsilon_{it} \tag{13}$$

$f_t = (f_{1t}, f_{2t}, f_{3t})'$ is the vector of common factors, $b_i = (b_{i1}, b_{i2}, b_{i3})'$ is the vector of factor loadings associated with the ith asset, and $\epsilon_{it}$ is the idiosyncratic component of $r_{it}$ satisfying $E(\epsilon_{it}|f_t) = 0$. We assume that $f_t \sim \mathcal{N}\left(\mu_f, \Sigma_f\right)$, where $\mu_f$ and $\Sigma_f$ are calibrated on the monthly data of the market portfolio, the Fama–French size, and the book-to-market portfolio from July 1980 to June 2016. Moreover, we assume that $b_i \sim \mathcal{N}(\mu_b, \Sigma_b)$ with $\mu_b$ and $\Sigma_b$ calibrated using data of 30 industry portfolios from July 1980 to June 2016. Idiosyncratic terms $\epsilon_{it}$ are supposed to be normally distributed. The covariance matrix of the residual vector is assumed to be diagonal and given by $\Sigma_\epsilon = \text{diag}(\sigma_1^2, ..., \sigma_N^2)$ with the diagonal elements drawn from a uniform distribution between 0.10 and 0.30 to yield an average cross-sectional volatility of 20%.

In the compact form (13) can be written as follows,

$$R = BF + \epsilon \tag{14}$$

where $B$ is a $N \times 3$ matrix whose ith row is $b_i'$. The covariance matrix of the vector of excess return $r_t$ is given by

$$\Sigma = B\Sigma_f B' + \Sigma_\epsilon.$$

The mean of the excess return is given by $\mu = B\mu_f$. The return on the risk-free asset $R_f$ is calibrated to be the mean of the one-month T-B observed in the data from July 1980 to June 2016.

The calibrated parameters used in our simulation process are given in Table 1. The gross return on the risk-free asset calibrated on the data is given by $R_f = 1.0036$. Once generated, the factor loadings are kept fixed over replications, while the factors differ from simulations and are drawn from a trivariate normal distribution.

**Table 1.** Calibrated parameters.

| Parameters for Factors Loadings | | | | Parameters for Factors Returns | | | |
|---|---|---|---|---|---|---|---|
| $\mu_b$ | | $\Sigma_b$ | | $\mu_f$ | | $\Sigma_f$ | |
| 1.0267 | 0.0422 | 0.0388 | 0.0115 | 0.0063 | 0.0020 | 0.0003 | −0.0004 |
| 0.0778 | 0.0388 | 0.0641 | 0.0162 | 0.0011 | 0.0003 | 0.0009 | −0.0003 |
| 0.2257 | 0.0115 | 0.0162 | 0.0862 | 0.0028 | −0.0004 | −0.0003 | 0.0009 |

Let $\text{SR}(\omega_t)$ be the Sharpe ratio associated with the optimal portfolio $\omega_t$, then $\text{SR}(\omega_t)$ is given as follows,

$$SR(\omega_t) = \left[\mu^{'}\Sigma\mu\right]^{1/2}$$

To evaluate the performance of our procedure in terms of the Sharpe ratio, we focus on the actual Sharpe ratio associated with the selected portfolio. The actual Sharpe ratio at time point $t$ is given by

$$SR(\hat{\omega}_t) = \frac{\hat{\omega}_t^{'}\mu}{\left[\hat{\omega}_t^{'}\Sigma\hat{\omega}_t^{'}\right]^{1/2}}$$

We consider the following portfolio selection procedures.

- The sample-based diversified portfolio (SbDP). This strategy is obtained using sample moments to estimate the unknown parameters in the maximum diversification portfolio.

$$SbDP = \frac{\hat{\Sigma}^{-1}\hat{\sigma}}{\hat{\sigma}^{'}\hat{\Sigma}^{-1}\hat{\sigma}}$$

- The most diversified portfolio (MDP) proposed by Choueifaty et al. (2013). This strategy is obtained by solving the optimization problem in Equation (2) under the following constraint,
  $\omega_i \geq 0 \; for \; i = 1, ..., N$.
  The closed form associated with this new optimization problem is given as follows,

$$MDP = diag(\Sigma)C^{-1}1$$

where $diag(\Sigma)$ is a diagonal matrix of assets volatilities, $C$ the correlation matrix, and 1 a $N \times 1$ vector of ones. The MDP is then estimated by replacing the unknown parameters by their empirical counterparts.
- The global minimum variance portfolio (GMVP) obtained by minimizing the variance of the return on the optimal selected portfolio. By solving this optimization problem, the following closed form is obtained,

$$GMVP = \frac{\Sigma^{-1}1}{1^{'}\Sigma^{-1}1}$$

This solution is then estimated by replacing the covariance matrix by the sample covariance matrix.
- The regularized strategies such as: the ridge regularized diversified portfolio (RdgDP), the spectral cut regularized diversified portfolio (SCDP), and the Landweber–Fridman regularized diversified portfolio (LFDP).
- The equal-weighted portfolio which is also called the naive portfolio (XoNP) which allocates a constant amount 1/N+1 in each asset.
- The target (or the maximum Sharpe ratio) portfolio (TgP). The closed form of the target portfolio is

$$TgP = \frac{\Sigma^{-1}\mu}{\mu^{'}\Sigma^{-1}1}$$

This portfolio is also estimated using sample moments such as the sample mean and the sample covariance matrix to estimate the unknown parameters.

- The linear factor-based shrinkage estimators proposed by Ledoit and Wolf (2003) (LWP). It consists of replacing the sample covariance matrix in the selected portfolio by an optimally weighted average of two existing estimators: the sample covariance matrix and single-index covariance matrix. This method involves also a tuning parameter that is unknown and has been selected by the authors. The tuning parameter selection procedure proposed in Ledoit and Wolf (2003) is based on minimizing the distance between the population covariance matrix and the regularized one. This implies that the way they select the turning parameter is different from our data-driven method. Therefore, the LWP will be considered here as a very good benchmark (and it will be the only benchmark that we consider) to evaluate the ability of our data-driven method to deliver additional performance compare to other data-driven methods.

We perform 1000 simulations and estimate our statistics over replications. We obtain the following results about the actual Sharpe ratio.

Table 2 contains the results about the average monthly Sharpe ratio obtained by simulations. The results show that the sample based diversified portfolio performs very poorly in terms of maximizing the Sharpe ratio in the financial market with large number of assets. This result is essentially due to the fact that the estimation error from estimating the vector of assets volatilities is amplified by using the sample covariance matrix of assets returns closed to a singular matrix when $N$ becomes too large compared with the sample size. Therefore, even if this strategy is supposed to be the maximum diversification's one with the highest Sharpe ratio, the SbDP is dominated by several other strategies such as the GMVP, the XoNP, and the TgP. Therefore, this strategy cannot be consider as the maximum diversification strategy in practice. To solve this problem, Choueifaty et al. (2013) proposes the most diversified portfolio (MDP) obtained by maximizing the diversification ratio under a non-negative constraint on the portfolio weights. This additional constraint in the investment process may help to reduce the effect of estimation error on the performance of the selected portfolio. The results of this analysis are in Table 2. By imposing the non-negative constraint, investors considerably improve the performance of the selected portfolio in terms of the Sharpe ratio. This new strategy even outperforms the global minimum variance portfolio. However, this procedure is still dominated by the target portfolio and the equal weighted portfolio meaning that much remains to be done about finding the maximum diversification strategy in practice. One explanation to this result is that imposing the non-negative constraint on the portfolio weight may limit the ability of the selected portfolio to be fully diversified. Therefore, one needs to find a more general estimation procedure for the maximum diversified portfolio that allows for short selling.

**Table 2.** The average monthly Actual Sharpe ratio from optimal strategies using a three-factor model as a function of the number of assets in the economy with the sample size $n = 120$, over 1000 replications. True SR is the true actual Sharpe ratio.

| Strategies | Number of Risky Assets | | | | | | |
|---|---|---|---|---|---|---|---|
| | 10 | 20 | 40 | 60 | 80 | 90 | 100 |
| SbDP | 0.1549 | 0.0906 | 0.0889 | 0.0779 | 0.0652 | 0.0719 | 0.0704 |
| XoNP | 0.2604 | 0.2604 | 0.2415 | 0.2525 | 0.2406 | 0.2461 | 0.2467 |
| GMVP | 0.2227 | 0.2338 | 0.2098 | 0.2298 | 0.1710 | 0.1640 | 0.1449 |
| MDP | 0.2514 | 0.2545 | 0.2410 | 0.2544 | 0.1778 | 0.1821 | 0.1935 |
| TgP | 0.2608 | 0.2818 | 0.2662 | 0.2687 | 0.2026 | 0.1925 | 0.1699 |
| LWP | 0.2589 | 0.2702 | 0.2688 | 0.2704 | 0.2628 | 0.2521 | 0.2507 |
| RdgDP | 0.2587 | 0.2785 | 0.2817 | 0.2907 | 0.2947 | 0.2830 | 0.2991 |
| SCDP | 0.2592 | 0.2872 | 0.2993 | 0.2898 | 0.2746 | 0.2887 | 0.2853 |
| LFDP | 0.2605 | 0.2765 | 0.2840 | 0.2870 | 0.2850 | 0.2912 | 0.2980 |
| True SR | 0.2626 | 0.2922 | 0.3393 | 0.3379 | 0.3592 | 0.3477 | 0.3657 |

For this purpose, we propose a new way to estimate the optimal diversified portfolio by stabilizing the inverse of the sample covariance matrix without imposing a non-negative constraint on the portfolio weights in the investment process. The results of these methods are also in Table 2. The first thing to point out about these results is that the regularized diversified portfolio outperforms the most diversified portfolio in terms of maximizing the Sharpe ratio. For instance, we obtain an average Sharpe ratio of 0.2514, 0.2587, 0.2592, and 0.2605 for the MDP, the RdgDP, the SCDP, and the LFDP, respectively, when only 10 assets are considered in the economy. The difference in terms of the actual Sharpe ratio performance between our procedure and the most diversified portfolio significantly increases with the number of assets in the financial market. For example, for 100 assets, the average Sharpe ratio is about 0.1935, 0.2991, 0.2853, and 0.2980 for the MDP, the RdgDP, the SCDP, and the LFDP, respectively. This results may be due to the fact that when the number of assets in the economy increases, the degree of diversification of the selected strategy may deteriorate with non-negative constraints on the investment process that may reduce the ability to find a strategy that performs the Sharpe ratio. Moreover, the regularized diversified portfolio outperforms the target strategy and the equal-weighted portfolio when the number of assets in the financial market exceeds 40. Nonetheless, for 10 assets in the economy, the target portfolio outperforms the RdgDP and the SCDP but is dominated by the LFDP. With 20 assets the target portfolio dominates the RdgDP and the LFDP and is dominated by the SCDP. The equal-weighted portfolio outperforms some regularized strategies such as the RdgDP and the SCDP only for 10 assets in the financial market. The fact that the regularized strategies give very interesting results in terms of maximizing the Sharpe ratio (compared with the existing strategies) for large $N$ is because these methods are essentially used to address estimation issues in large dimensional problems. The performance of these procedures seems to be independent of the size of the financial market. In fact, with a reasonable choice of the tuning parameter, each of these methods can achieve satisfactory performance in terms of the Sharpe ratio even if the number of assets in the economy is large.

Our regularized portfolio also outperforms the selected strategy obtained using the linear shrinkage estimator of Ledoit and Wolf (2003) to estimate the covariance matrix of asset returns. The difference in terms of performance between these two portfolios tends to become large when the number of assets we consider in the economy increases. This result can be due to the fact that the estimation error associated with estimating the single-index covariance matrix may be important for very large assets. One other thing that could explain this result comes from the fact that our tuning parameter is selected to minimize the distance between the regularized inverse of the covariance matrix and the inverse of the population's one. Moreover, because the optimal portfolio depends on the inverse of the covariance matrix, selecting a tuning parameter that minimizes the estimation error in the inverse of the covariance matrix seems to be more appropriate than choosing this parameter to minimize the estimation error in the covariance matrix. One important thing to point out is that the Ridge regularized portfolio is a special case of the selected portfolio with the linear shrinkage estimation of the covariance matrix. In this case, the structural covariance matrix is replaced by the identity to avoid the potential estimation error which may be associated with this covariance matrix.

Similar results are obtained when choosing the estimation window to be 1000 and by increasing the number of assets in the economy from 150 to 999 ($N \in \{150, 250, 400, 550, 700, 850, 950, 999\}$) as given in Table 3.

**Table 3.** The average monthly Actual Sharpe ratio from optimal strategies using a three-factor model as a function of the number of assets in the economy with the sample size $n = 1000$, over 1000 replications. True SR is the true actual Sharpe ratio.

| Strategies | Number of Risky Assets | | | | | | | |
|---|---|---|---|---|---|---|---|---|
| | 150 | 250 | 400 | 550 | 700 | 850 | 950 | 999 |
| SbDP | 0.1230 | 0.1104 | 0.103 | 0.0998 | 0.060 | 0.03 | 0.012 | 0.008 |
| XoNP | 0.2630 | 0.2640 | 2507 | 0.240 | 0.238 | 0.2207 | 0.2180 | 0.220 |
| GMVP | 0.3080 | 02908 | 0.2890 | 0.2780 | 0.250 | 0.1980 | 0.1017 | 0.095 |
| MDP | 0.3280 | 0.3305 | 0.3198 | 0.309 | 0.2679 | 0.2892 | 0.1985 | 0.120 |
| TgP | 0.3290 | 0.3105 | 0.307 | 0.3100 | 0.2608 | 0.210 | 0.180 | 0.098 |
| LWP | 0.3302 | 0.3408 | 0.3318 | 0.3070 | 0.415 | 0.4504 | 0.4601 | 0.4807 |
| RdgDP | 0.3702 | 0.3850 | 0.3980 | 0.458 | 0.524 | 0.540 | 0.558 | 0.601 |
| SCDP | 0.3689 | 0.3860 | 0.3980 | 0.460 | 0.5230 | 0.535 | 0.590 | 0.608 |
| LFDP | 0.3704 | 0.3840 | 0.3984 | 0.4560 | 0.5250 | 0.538 | 0.585 | 0.595 |
| True SR | 0.3758 | 0.3904 | 0.407 | 0.489 | 0.5480 | 0.588 | 0.608 | 0.618 |

To analyze the statistical significance of the regularized portfolio over the other strategies, we implement the following test procedure about the Sharpe ratio,

$$H_0 : RSR \leq SR_0 \, vs \, H_1 : RSR > SR_0$$

where $RSR$ is the regularized Sharpe ratio and $SR_0$ the Sharpe ratio of the portfolio under comparison. This test is conducted using the same procedure as in Ao et al. (2019). For more information about this test procedure see Jobson and Korkie (1981) and Memmel (2003). The fundamental objective of this test procedure is to confirm the domination of our method over the existing strategies with a statistic test. The *p*-values associated with this test procedure for each of the regularized portfolios are given in Tables 4–6. According to these results, our regularized portfolio dominates the other strategies in terms of maximizing the Sharpe ratio at the significant level 5%. In particular, our method outperforms the LW portfolio in the large financial market setting.

**Table 4.** The *p*-value associated with performance hypothesis testing with the Sharpe ratio from Ridge regularized strategy using a three-factor model as a function of the number of assets in the economy with the sample size $n = 1000$, over 1000 replications.

| Strategies | Number of Risky Assets | | | | | | | |
|---|---|---|---|---|---|---|---|---|
| | 150 | 250 | 400 | 550 | 700 | 850 | 950 | 999 |
| SbDP | 0.000 | 0.000 | 0.000 | 0.000 | 0.000 | 0.000 | 0.000 | 0.000 |
| XoNP | 0.004 | 0.002 | 0.007 | 0.005 | 0.000 | 0.000 | 0.000 | 0.000 |
| GMVP | 0.008 | 0.004 | 0.006 | 0.007 | 0.000 | 0.000 | 0.000 | 0.000 |
| MDP | 0.003 | 0.001 | 0.002 | 0.000 | 0.000 | 0.000 | 0.000 | 0.000 |
| TgP | 0.009 | 0.003 | 0.008 | 0.004 | 0.001 | 0.000 | 0.008 | 0.000 |
| LWP | 0.089 | 0.013 | 0.001 | 0.012 | 0.035 | 0.003 | 0.043 | 0.008 |

**Table 5.** The *p*-value associated with Performance hypothesis testing with the Sharpe ratio from Landweber–Fridman regularized strategy using a three-factor model as a function of the number of assets in the economy with the sample size $n = 1000$, over 1000 replications.

| Strategies | Number of Risky Assets | | | | | | | |
|---|---|---|---|---|---|---|---|---|
| | 150 | 250 | 400 | 550 | 700 | 850 | 950 | 999 |
| SbDP | 0.000 | 0.000 | 0.000 | 0.000 | 0.000 | 0.000 | 0.000 | 0.000 |
| XoNP | 0.003 | 0.001 | 0.008 | 0.007 | 0.001 | 0.000 | 0.000 | 0.000 |
| GMVP | 0.010 | 0.003 | 0.007 | 0.002 | 0.001 | 0.000 | 0.000 | 0.000 |
| MDP | 0.005 | 0.001 | 0.004 | 0.000 | 0.000 | 0.000 | 0.000 | 0.000 |
| TgP | 0.008 | 0.004 | 0.005 | 0.004 | 0.002 | 0.000 | 0.008 | 0.000 |
| LWP | 0.090 | 0.014 | 0.003 | 0.009 | 0.040 | 0.007 | 0.001 | 0.007 |

**Table 6.** The *p*-value associated with performance hypothesis testing with the Sharpe ratio from spectral cut-off regularized strategy using a three-factor model as a function of the number of assets in the economy with the sample size $n = 1000$, over 1000 replications.

| Strategies | Number of Risky Assets | | | | | | | |
|---|---|---|---|---|---|---|---|---|
| | 150 | 250 | 400 | 550 | 700 | 850 | 950 | 999 |
| SbDP | 0.000 | 0.000 | 0.000 | 0.000 | 0.000 | 0.000 | 0.000 | 0.000 |
| XoNP | 0.004 | 0.003 | 0.006 | 0.005 | 0.000 | 0.000 | 0.000 | 0.000 |
| GMVP | 0.020 | 0.003 | 0.005 | 0.001 | 0.000 | 0.000 | 0.000 | 0.000 |
| MDP | 0.003 | 0.002 | 0.003 | 0.002 | 0.001 | 0.000 | 0.000 | 0.000 |
| TgP | 0.003 | 0.002 | 0.004 | 0.002 | 0.001 | 0.000 | 0.001 | 0.000 |
| LWP | 0.104 | 0.043 | 0.002 | 0.008 | 0.032 | 0.004 | 0.002 | 0.006 |

We also compute in Table 7 the average monthly diversification ratio associated with the selected portfolio. We obtain similar results to what has been obtained in Table 2. The regularized portfolio performs well in terms of maximizing the diversification ratio and dominated most of the existing methods in the large financial market. The diversification ratio that we obtain with our method is very close to the true one. This implies that in addition to the asymptotic results obtained in the Section 4, the regularized portfolio has very good finite sample properties. This result shows that we do not need $N/\sqrt{T}$ to be close to zero to improve the finite sample performance of the selected portfolio.

**Table 7.** The average monthly Actual monthly diversification ratio from optimal strategies using a three-factor model as a function of the number of assets in the economy with the sample size $n = 120$, over 1000 replications. True DR is the true diversification ratio.

| Strategies | Number of Risky Assets | | | | | | |
|---|---|---|---|---|---|---|---|
| | 10 | 20 | 40 | 60 | 80 | 90 | 100 |
| SbDP | 2.315 | 2.307 | 2.304 | 2.08 | 1.308 | 1.128 | 1.098 |
| XoNP | 3.103 | 3.140 | 3.180 | 3.184 | 3.325 | 3.288 | 3.154 |
| GMVP | 3.242 | 3.241 | 3.150 | 3.185 | 3.147 | 3.155 | 3.093 |
| MDP | 3.252 | 3.320 | 3.240 | 3.290 | 3.320 | 3.265 | 3.254 |
| TgP | 3.240 | 3.170 | 3.105 | 3.050 | 3.132 | 3.149 | 3.080 |
| LWP | 3.345 | 3.360 | 3.320 | 3.380 | 3.398 | 3.403 | 3.420 |
| RdgDP | 3.325 | 3.428 | 3.480 | 3.590 | 3.598 | 3.602 | 3.640 |
| SCDP | 3.347 | 3.435 | 3.446 | 3.570 | 3.589 | 3.615 | 3.625 |
| LFDP | 3.289 | 3.405 | 3.470 | 3.548 | 3.604 | 3.509 | 3.638 |
| True DR | 3.45 | 3.56 | 3.57 | 3.68 | 3.8 | 3.7 | 3.9 |

## 5.3. Empirical Study

In this empirical section, our objective is to use the real data (unlike in the simulation part) to estimate the unknown parameters of the optimal portfolio and then to evaluate the

performance of each estimation procedure based on the same statistics as in the simulation section. Note that our purpose in this paper lies not in forecasting but proposing a consistent way that allows us to correctly estimate the portfolio in Equation (4) in large dimensional setting.

We apply our method to several sets of portfolios from Kenneth R. French's website. In particular, we apply our procedure to the following portfolios: the 30-industry portfolios and the 100 portfolios formed on size and book-to-market. We allow investors to rebalance their portfolios every month. This implies that the optimal portfolio is constructed at the end of each month for a given estimation window *M* by maximizing the diversification ratio. The investor holds this portfolio for one month, realizes gains and losses, updates information, and then recomputes optimal portfolio weights for the next period using the same estimation window. This procedure is repeated each month, generating a time series of out-of-sample returns. This time series can then be used to analyze the out-of-sample performance of each strategy based on several statistics such as the out-of-sample Sharpe ratio. For this purpose, we use data from July 1980 to June 2018.

Table 8 contains some results of the out-of-sample analysis in terms of the Sharpe ratio for two different data sets: the FF30 and the FF100. The empirical results in this table confirm what we have obtained in the simulation part. According to this result, by stabilizing the inverse of the covariance matrix in the maximum diversification portfolio, we considerably improve the performance of the selected strategy in terms of maximizing the Sharpe ratio. Moreover, our regularized strategies outperform the most diversified strategy, the target portfolio, The LW portfolio, and the global minimum variance portfolio for each data set. The most diversified strategy outperforms the global minimum variance portfolio but is dominated by the Equal-Weight portfolio for each data set. These results of the most diversified portfolio can essentially be explained by the fact that by imposing a non-negative constraint in the investment process, one cannot fully diversify the optimal portfolio. The LWP outperforms the other strategies, in particular, this method dominates the most diversified strategy of Choueifaty et al. (2013). The return of the regularized portfolio is less volatile than what we obtain with the most diversified portfolio, the target one, and the LW strategy.

**Table 8.** Out-of-sample performance in terms of the Sharpe ratio applied on the 30 industry portfolios (FF30) and the 100 portfolios formed on size and book-to-market (FF100) with a rolling window of 120.

| **Strategies** | | **XoNP** | **GMVP** | **MDP** | **TGP** | **RdgP** | **LFP** | **SCP** | **LWP** |
|---|---|---|---|---|---|---|---|---|---|
| | ER | 0.0110 | 0.01134 | 0.0121 | 0.017 | 0.0149 | 0.014 | 0.014 | 0.014 |
| FF30 | V | 0.0540 | 0.0630 | 0.058 | 0.076 | 0.063 | 0.057 | 0.061 | 0.067 |
| | **SR** | 0.204 | 0.180 | 0.209 | 0.224 | 0.237 | 0.246 | 0.2295 | 0.209 |
| | ER | 0.0103 | 0.0127 | 0.015 | 0.0173 | 0.0200 | 0.0201 | 0.0203 | 0.019 |
| FF100 | V | 0.0485 | 0.075 | 0.088 | 0.091 | 0.0772 | 0.0770 | 0.078 | 0.082 |
| | **SR** | 0.212 | 0.1693 | 0.1705 | 0.1901 | 0.2590 | 0.2610 | 0.2602 | 0.2317 |

We are also interested in how our procedure can perform in terms of minimizing the rebalancing cost at a given period. The rebalancing cost at the time *t* can be naturally measured by

$$Cost_t = \sum_{j=1}^{N} \left| \omega_{t,j} - \omega_{t-1,j} \right|.$$

This measure of the trading cost is, in fact, the turnover. The transaction cost can be measured using the turnover in the sense that these costs are positively related to the

turnover. Therefore, in the rest of the paper the turnover will be called transaction costs. The average trading cost over the investment horizon is given by

$$TradingCost = \frac{1}{Q} \sum_{t=1}^{Q} Cost_t$$

where $Q$ is the number of rebalancing periods. This quantity can be interpreted as the average percentage of wealth traded at each period. The average monthly rebalancing costs are given in Table 9. These results show that by stabilizing the inverse of the covariance matrix by regularization, we help investors to select strategies that significantly reduce the rebalancing cost. The regularized portfolio outperforms the other strategies in terms of minimizing the trading costs faced by investors in their investment process.

**Table 9.** Out-of-sample performance in terms of rebalancing cost (turnover) applied on the 30 industry portfolios (FF30) and the 100 portfolios formed on size and book-to-market (FF100) for two different rolling windows.

| P | EW | Strategies | | | | | | | |
|---|---|---|---|---|---|---|---|---|---|
| | | **SbDP** | **GMVP** | **MDP** | **TgP** | **LWP** | **RdgDP** | **SCDP** | **LFDP** |
| FF30 | 60 | 6.890 | 4.329 | 2.809 | 4.209 | 1.0328 | 0.9952 | 0.989 | 0.9872 |
| | 120 | 5.605 | 3.901 | 2.087 | 3.290 | 0.9892 | 0.7140 | 0.7203 | 0.6450 |
| FF100 | 120 | 9.789 | 6.2390 | 5.978 | 6.309 | 1.7808 | 1.3267 | 1.3890 | 1.2078 |
| | 240 | 7.089 | 4.297 | 3.879 | 4.2870 | 1.3065 | 1.0349 | 1.0398 | 1.096 |

The evolution of the share of the selected assets in the optimal portfolio in Figure 1 shows that by regularizing the covariance matrix, we considerably reduce extreme positions in the selected strategy. Therefore, we significantly reduce the transaction costs faced by investors when they decide to take positions in the financial market. Moreover, the return on the selected portfolio becomes less volatile in such a situation.

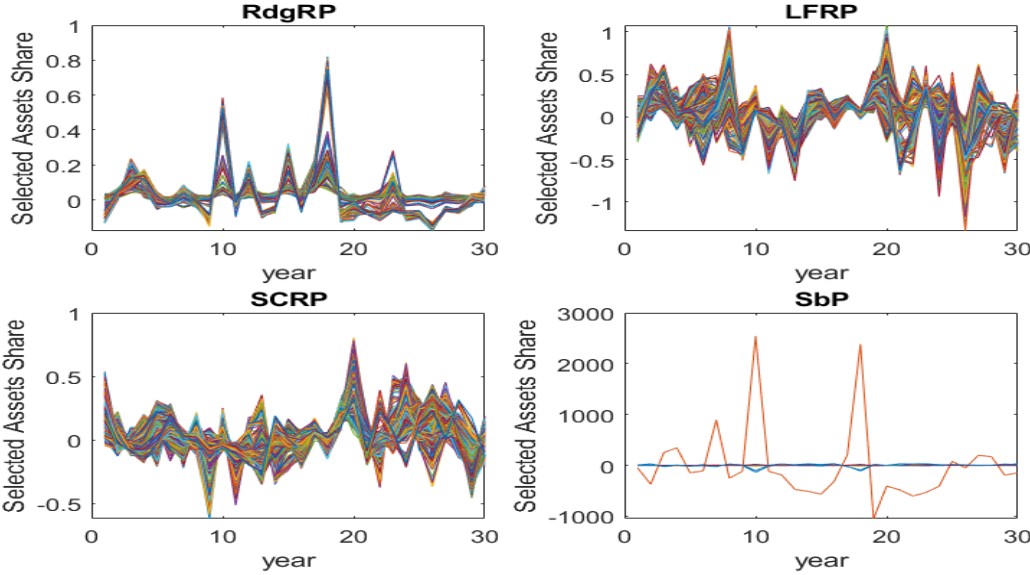

**Figure 1.** The evolution of the selected assets in the optimal portfolio. We obtain this figure using the 30 industry portfolios with an estimation window of $n = 120$.

Tables 10 and 11 contain the Fama–French monthly regression coefficients for the 100 portfolios formed on size and book-to-market and the 30-industry portfolios, respectively. Monthly data are used from July 1990 to June 2018. According to the result in

Table 10, only the return on the Equal-Weight portfolio can be explained by the Fama–French three-factor model for the 100 portfolios formed on size and book-to-market. The return obtained with the regularized portfolios and the most diversified portfolio can be explained only with the return on the market portfolio (a one-factor model) through a positive relation. However, the return of the most diversified portfolio and the global minimum variance portfolio can be explained with a two factors model when using the 30-industry portfolios. The return of the other strategies such as the regularized portfolios, the Equal-Weight portfolio, and the target portfolio can be explained by the Fama–French three-factor model.

**Table 10.** Fama–French Monthly Regression Coefficients for the 100 portfolios formed on size and book-to-market from July 1990 to June 2018.

| Strategies | Market | HML | SMB | Intercept |
|---|---|---|---|---|
| Rdg-regularized Portfolio | 0.9168 (0.000) | 0.079 (0.531) | −0.139 (0.302) | 0.0075 (0.057) |
| LF- regularized Portfolio | 0.823 (0.000) | 0.174 (0.153) | −0.1651 (0.204) | 0.0125 (0.001) |
| SC-regularized Portfolio | 1.02 (0.000) | −0.127 (0.177) | −0.133 (0.189) | 0.0077 (0.010) |
| Most-Diversified Portfolio | 0.72 (0.000) | 0.13 (0.344) | 0.098 (0.506) | 0.007 (0.002) |
| Equal-Weight-Portfolio | 1.002 (0.000) | 0.5104 (0.000) | 0.33 (0.000) | 0.0001 (0.815) |
| Global-Minimum-Variance Portfolio | 0.416 (0.000) | −0.125 (0.319) | 0.155 (0.247) | 0.0094 (0.000) |
| Target-Portfolio | 0.43 (0.000) | 0.144 (0.367) | 0,207 (0.226) | 0.010 (0.000) |
| LW-Portfolio | 0.802 (0.000) | 0.074 (0.247) | 0.207 (0.226) | 0.0082 (0.067) |

**Table 11.** Fama–French Monthly Regression Coefficients for the 30-industry portfolios from July 1990 to June 2018.

| Strategies | Market | HML | SMB | Intercept |
|---|---|---|---|---|
| Rdg-regularized Portfolio | 1.03 (0.000) | 0.24 (0.003) | 0.36 (0.000) | 0.0007 (0.767) |
| LF- regularized Portfolio | 0.93 (0.000) | 0.22 (0.003) | 0.25 (0.001) | 0.0046 (0.042) |
| SC-regularized Portfolio | 0.86 (0.000) | 0.27 (0.000) | 0.21 (0.031) | 0.0054 (0.053) |
| Most-Diversified Portfolio | 0.46 (0.000) | −0.285 (0.000) | 0.070 (0.391) | 0.002 (0.001) |
| Equal-Weight-Portfolio | 0.983 (0.000) | 0.061 (0.006) | 0.265 (0.000) | 0.0013 (0.050) |
| Global-Minimum-Variance Portfolio | 0.46 (0.000) | −0.146 (0.008) | 0.077 (0.188) | 0.0021 (0.017) |
| Target-Portfolio | 0.54 (0.000) | −0.44 (0.000) | −0.21 (0.019) | 0.013 (0.000) |
| LW-Portfolio | 0.982 (0.000) | 0.272 (0.0098) | 0.4112 (0.0301) | 0.0006 (0.429) |

As the portfolio optimization is generally based on individual stocks instead of aggregate portfolios as the Fama–French portfolio, we apply also our method to a subset of the S&P500 index constituents to see how our method performs in such universe. We use for this purpose monthly data from March 1986 to December 2019. At the beginning of this empirical analysis, we randomly form pools of 100 or 150 stocks from the S&P500 index constituents for which there are complete return data for the prior 120 or 240 months. The

optimal portfolio will then be constructed using the same procedure as before. We then compute the out-of-sample performance in terms of the Sharpe ratio and the turnover. The results of this empirical analysis are given in Tables 12 and 13. We obtain similar results as in the case of the Fama–French portfolios proving that our method also performs well when the optimal portfolio is formed with individual stocks from S&P500.

**Table 12.** Out-of-sample performance in terms of Sharpe ratio applied on two subsets of S&P500 constituents for two different rolling windows.

| P | EW | Strategies | | | | | | | |
| | | SbDP | GMVP | MDP | TgP | LWP | RdgDP | SCDP | LFDP |
|---|---|---|---|---|---|---|---|---|---|
| 100 A | 120 | 0.0850 | 0.1506 | 0.2458 | 0.1983 | 0.3702 | 0.4382 | 0.4380 | 0.4397 |
| | 240 | 0.0982 | 0.1604 | 0.260 | 0.2028 | 0.3809 | 0.4565 | 0.4567 | 0.4578 |
| 150 A | 180 | 0.0750 | 0.1204 | 0.309 | 0.1407 | 0.4108 | 0.5353 | 0.5320 | 0.5462 |
| | 240 | 0.0895 | 0.1750 | 0.320 | 0.1890 | 0.4208 | 0.5603 | 0.5609 | 0.5579 |

**Table 13.** Out-of-sample performance in terms of rebalancing cost (turnover) applied on two subsets of S&P500 constituents for two different rolling windows.

| Assets | EW | Strategies | | | | | | | |
| | | SbDP | GMVP | MDP | TgP | LWP | RdgDP | SCDP | LFDP |
|---|---|---|---|---|---|---|---|---|---|
| 100 Assets | 120 | 9.450 | 6.786 | 4.675 | 6.679 | 3.348 | 2.1067 | 2.0801 | 2.0682 |
| | 240 | 6.978 | 5.308 | 3.892 | 5.234 | 3.078 | 1.491 | 1.608 | 1.569 |
| 150 Assets | 180 | 10.489 | 7.345 | 6.782 | 7.328 | 3.897 | 2.678 | 2.780 | 2.8960 |
| | 240 | 8.0789 | 5.542 | 4.032 | 5.438 | 3.057 | 2.104 | 2.0978 | 2.0956 |

## 6. Conclusions

This paper addresses the estimation issue that exists in the maximum diversification portfolio framework in the large financial market. We propose to stabilize the inverse of the covariance matrix in the diversified portfolio using regularization techniques from inverse problem literature. These regularization techniques, namely, the ridge, the spectral cut-off, and Landweber–Fridman, involve a regularization parameter or penalty term whose optimal value is selected to minimize the expected distance between the inverse of the estimated covariance matrix and the inverse of the true covariance matrix. We show, under appropriate regularity conditions, that the selected strategy by regularization is asymptotically efficient with respect to the diversification ratio for a wise choice of the tuning parameter. Meaning that, even if the diversified portfolio is unknown, there exists a feasible portfolio obtained by regularization capable of reaching a similar level of performance in terms of the diversification ratio.

To evaluate the performance of our procedures, we implement a simulation exercise based on a three-factor model calibrated on real data from the US financial market. We obtain by simulation that our procedure significantly improves the performance of the selected strategy with respect to the Sharpe ratio. Moreover, the regularized rules are compared to several strategies such as the most diversified portfolio, the target portfolio, the global minimum variance portfolio, and the naive 1/N strategy in terms of in-sample and out-of-sample Sharpe ratio, and it is shown that our method yields significant Sharpe ratio improvements. To confirm our simulations, we do an empirical analysis using Kenneth R. French's 30-industry portfolios and 100 portfolios formed on size and book-to-market. According to this empirical result, by stabilizing the inverse of the covariance matrix in the maximum diversification portfolio, we considerably improve the performance of the selected strategy in terms of maximizing the Sharpe ratio.

**Funding:** This research received no external funding.

**Institutional Review Board Statement:** Not applicable.

**Informed Consent Statement:** Not applicable.

**Data Availability Statement:** The data presented in this study are available on request from the corresponding author.

**Conflicts of Interest:** The authors declare no conflict of interest.

## Appendix A. Proof of Proposition 1

By definition we have that

$$DR(\hat{\omega}_\alpha) = \frac{\hat{\omega}_\alpha' \sigma}{\sqrt{\hat{\omega}_\alpha' \Sigma \hat{\omega}_\alpha}}.$$

Let us first look at $\hat{\omega}_\alpha' \Sigma \hat{\omega}_\alpha$

$$
\begin{aligned}
\hat{\omega}_\alpha' \Sigma \hat{\omega}_\alpha &= [(\hat{\omega}_\alpha - \omega) + \omega]' \Sigma [(\hat{\omega}_\alpha - \omega) + \omega] \\
&= \omega' \Sigma \omega + \underbrace{(\hat{\omega}_\alpha - \omega)' \Sigma (\hat{\omega}_\alpha - \omega)}_{(a)} + 2 \underbrace{(\hat{\omega}_\alpha - \omega)' \Sigma \omega}_{(b)}.
\end{aligned}
$$

Now we are going to look at the properties of (a) and (b). We know that

$$\hat{\omega}_\alpha = \left( \underbrace{\hat{\sigma}' \hat{\Sigma}^\alpha \hat{\sigma}}_{(c)} \right)^{-1} \underbrace{\hat{\Sigma}^\alpha \hat{\sigma}}_{(d)}.$$

$$
\begin{aligned}
(c) &= \sigma' \hat{\Sigma}^\alpha \sigma + (\hat{\sigma} - \sigma)' \hat{\Sigma}^\alpha (\hat{\sigma} - \sigma) + 2(\hat{\sigma} - \sigma)' \hat{\Sigma}^\alpha \sigma \\
\hat{\Sigma}^\alpha &= \left( \hat{\Sigma}^\alpha - \Sigma^\alpha + \Sigma^\alpha \right).
\end{aligned}
$$

$$
\begin{aligned}
\left\| (\hat{\sigma} - \sigma)' \hat{\Sigma}^\alpha (\hat{\sigma} - \sigma) \right\| &= \left\| \frac{(\hat{\sigma} - \sigma)'}{\sqrt{N}} \left( \frac{\hat{\Sigma}}{N} \right)^\alpha \frac{(\hat{\sigma} - \sigma)}{\sqrt{N}} \right\| \\
&= O_p \left( \frac{\|\hat{\sigma} - \sigma\|^2}{N\alpha} \right) \\
&= O_p \left( \frac{\left\| \frac{\hat{\sigma} - \sigma}{\sqrt{N}} \right\|^2}{\alpha} \right).
\end{aligned}
$$

By Assumption A $\left\| \frac{\sigma}{\sqrt{N}} \right\| = O(1)$. Therefore, we obtain that

$$
\begin{aligned}
\left\| (\hat{\sigma} - \sigma)' \hat{\Sigma}^\alpha \sigma \right\| &= \left\| \frac{(\hat{\sigma} - \sigma)'}{\sqrt{N}} \left( \frac{\hat{\Sigma}}{N} \right)^\alpha \frac{\sigma}{\sqrt{N}} \right\| \\
&= O_p \left( \frac{\|\hat{\sigma} - \sigma\|}{\sqrt{N}\alpha} \right) \\
&= O_p \left( \frac{\left\| \frac{\hat{\sigma} - \sigma}{\sqrt{N}} \right\|}{\alpha} \right).
\end{aligned}
$$

Using those information combine with the fact that $\hat{\Sigma}^\alpha = \hat{\Sigma}^\alpha - \Sigma^\alpha + \Sigma^\alpha$, we have that

$$(c) = \sigma'\Sigma^\alpha\sigma + \sigma'\left(\hat{\Sigma}^\alpha - \Sigma^\alpha\right)\sigma + O_p\left(\frac{\left\|\frac{\hat{\sigma}-\sigma}{\sqrt{N}}\right\| + \left\|\frac{\hat{\sigma}-\sigma}{\sqrt{N}}\right\|^2}{\alpha}\right).$$

$$\left\|\sigma'\left(\hat{\Sigma}^\alpha - \Sigma^\alpha\right)\sigma\right\| = \left\|\frac{\sigma'}{\sqrt{N}}\left[\left(\frac{\hat{\Sigma}}{N}\right)^\alpha - \left(\frac{\Sigma}{N}\right)^\alpha\right]\frac{\sigma}{\sqrt{N}}\right\|$$

$$\leq \left\|\frac{\sigma}{\sqrt{N}}\right\|^2 \left\|\left(\frac{\hat{\Sigma}}{N}\right)^\alpha - \left(\frac{\Sigma}{N}\right)^\alpha\right\|$$

$$= O_p\left(\left\|\left(\frac{\hat{\Sigma}}{N}\right)^\alpha - \left(\frac{\Sigma}{N}\right)^\alpha\right\|\right).$$

$$\left\|\left(\frac{\hat{\Sigma}}{N}\right)^\alpha - \left(\frac{\Sigma}{N}\right)^\alpha\right\| \leq \left\|\left(\frac{\Sigma}{N}\right)^\alpha\right\|\left\|\left(\frac{\hat{\Sigma}}{N}\right)^\alpha\right\|\left\|\frac{\hat{\Sigma}}{N} - \frac{\Sigma}{N}\right\|.$$

Hence,

$$\left\|\left(\frac{\hat{\Sigma}}{N}\right)^\alpha - \left(\frac{\Sigma}{N}\right)^\alpha\right\| = O_p\left(\frac{\left\|\frac{\hat{\Sigma}}{N} - \frac{\Sigma}{N}\right\|}{\alpha}\right)$$

which implies that

$$(c) = \sigma'\Sigma^\alpha\sigma + O_p\left(\frac{\left\|\frac{\hat{\Sigma}}{N} - \frac{\Sigma}{N}\right\| + \left\|\frac{\hat{\sigma}-\sigma}{\sqrt{N}}\right\| + \left\|\frac{\hat{\sigma}-\sigma}{\sqrt{N}}\right\|^2}{\alpha}\right).$$

As $T \to \infty$ we have that $\alpha \to 0 \Rightarrow$

$$(c) = \sigma'\Sigma^{-1}\sigma + O_p\left(\frac{\left\|\frac{\hat{\Sigma}}{N} - \frac{\Sigma}{N}\right\| + \left\|\frac{\hat{\sigma}-\sigma}{\sqrt{N}}\right\| + \left\|\frac{\hat{\sigma}-\sigma}{\sqrt{N}}\right\|^2}{\alpha}\right).$$

Using Assumption A combined with Theorem 4 of Carrasco and Florens (2000), we have that

$$\left\|\frac{\hat{\Sigma}}{N} - \frac{\Sigma}{N}\right\| = O_p\left(\frac{1}{\sqrt{T}}\right).$$

Moreover, as $\left\|\frac{\hat{\sigma}-\sigma}{\sqrt{N}}\right\|^2 = O_p\left(\frac{1}{T}\right)$ by Assumption A, we have that

$$(c) = \sigma'\Sigma^{-1}\sigma + O_p\left(\frac{1}{\alpha\sqrt{T}}\right).$$

$$\begin{aligned}
(d) &= \hat{\Sigma}^\alpha\hat{\sigma} \\
&= \hat{\Sigma}^\alpha\sigma + \hat{\Sigma}^\alpha(\hat{\sigma} - \sigma) \\
&= \Sigma^\alpha\sigma + \left(\hat{\Sigma}^\alpha - \Sigma^\alpha\right)\sigma + \hat{\Sigma}^\alpha(\hat{\sigma} - \sigma).
\end{aligned}$$

As $\alpha \to 0$ as $T \to \infty$, we have that

$$(d) = \Sigma^{-1}\sigma + (\hat{\Sigma}^{\alpha} - \Sigma)\sigma + \hat{\Sigma}^{\alpha}(\hat{\sigma} - \sigma).$$

We know that

$$
\begin{aligned}
\left\|\hat{\Sigma}^{\alpha}(\hat{\sigma} - \sigma)\right\| &= \left\|\left(\frac{\hat{\Sigma}}{N}\right)^{\alpha}\frac{(\hat{\sigma} - \sigma)}{N}\right\| \\
&\leq \left\|\left(\frac{\hat{\Sigma}}{N}\right)^{\alpha}\right\|\left\|\frac{(\hat{\sigma} - \sigma)}{N}\right\| \\
&= O_p\left(\frac{1}{\alpha\sqrt{TN}}\right).
\end{aligned}
$$

Using the fact that

$$
\begin{aligned}
\left\|(\hat{\Sigma}^{\alpha} - \Sigma)\sigma\right\| &= \left\|\left\{\left(\frac{\hat{\Sigma}}{N}\right)^{\alpha} - \left(\frac{\Sigma}{N}\right)^{\alpha}\right\}\frac{\sigma}{N}\right\| \\
&\leq \left\|\left(\frac{\hat{\Sigma}}{N}\right)^{\alpha} - \left(\frac{\Sigma}{N}\right)^{\alpha}\right\|\left\|\frac{\sigma}{N}\right\| \\
&= O_p\left(\frac{\left\|\frac{\hat{\Sigma}}{N} - \frac{\Sigma}{N}\right\|}{\alpha\sqrt{N}}\right) \\
&= O_p\left(\frac{1}{\alpha\sqrt{TN}}\right)
\end{aligned}
$$

we obtain that

$$(d) = \Sigma^{-1}\sigma + O_p\left(\frac{1}{\alpha\sqrt{TN}}\right).$$

Under the assumption that $\frac{1}{\alpha\sqrt{T}} \to 0$, we have that

$$\hat{\omega}_{\alpha} = \omega + o_p(1). \tag{A1}$$

By Assumption A we have that $\|\Sigma\| = O(N)$. Therefore, using (A1), we obtain that

$$\hat{\omega}'_{\alpha}\Sigma\hat{\omega}_{\alpha} = \omega'\Sigma\omega + o_p(1) \tag{A2}$$

if $\frac{N}{\alpha\sqrt{T}} \to 0$. Therefore,

$$DR(\hat{\omega}_{\alpha}) \to_p DR(\omega_t).$$

**Appendix B. Proof of Proposition 2**

$$(A) = \mu'\left[\left(\hat{\Sigma}^{\alpha} - \Sigma^{-1}\right)'\Sigma\left(\hat{\Sigma}^{\alpha} - \Sigma^{-1}\right)\right]\mu$$

We also know that $\mu = \hat{\mu} + (\mu - \hat{\mu})$, so

$$
\begin{aligned}
(A) \;&=\; \mu'\left[\left(\hat{\Sigma}^{\alpha} - \Sigma^{-1}\right)'\Sigma\left(\hat{\Sigma}^{\alpha} - \Sigma^{-1}\right)\right]\mu \\[4pt]
&=\; \left[\hat{\Sigma}^{\alpha}(\mu - \hat{\mu}) + \left(\hat{\Sigma}^{\alpha}\hat{\mu} - \Sigma^{-1}\mu\right)\right]'\Sigma\left[\hat{\Sigma}^{\alpha}(\mu - \hat{\mu}) + \left(\hat{\Sigma}^{\alpha}\hat{\mu} - \Sigma^{-1}\mu\right)\right] \\[4pt]
&=\; \left(\hat{\Sigma}^{\alpha}\hat{\mu} - \Sigma^{-1}\mu\right)'\Sigma\left(\hat{\Sigma}^{\alpha}\hat{\mu} - \Sigma^{-1}\mu\right) + \left[\hat{\Sigma}^{\alpha}(\mu - \hat{\mu})\right]'\Sigma\left[\hat{\Sigma}^{\alpha}(\mu - \hat{\mu})\right] \\[4pt]
&+\; 2\left[\hat{\Sigma}^{\alpha}(\mu - \hat{\mu})\right]'\Sigma\left(\hat{\Sigma}^{\alpha}\hat{\mu} - \Sigma^{-1}\mu\right)
\end{aligned}
$$

Let denote by $x = \Sigma^{-1}\mu$ and $\hat{x} = \hat{\Sigma}^{\alpha}\hat{\mu}$; therefore,

$$
(A) = (\hat{x} - x)'\Sigma(\hat{x} - x) + \left[\hat{\Sigma}^{\alpha}(\mu - \hat{\mu})\right]'\Sigma\left[\hat{\Sigma}^{\alpha}(\mu - \hat{\mu})\right] + 2\left[\hat{\Sigma}^{\alpha}(\mu - \hat{\mu})\right]'\Sigma(\hat{x} - x)
$$

As $\|\mu - \hat{\mu}\|^2 = O_p\left(\frac{N}{T}\right)$, $\left\|\left(\frac{\hat{\Sigma}}{N}\right)^{\alpha}\right\|^2 = O_p\left(\frac{1}{\alpha^2}\right)$, we have that

$$
\begin{aligned}
\left\|\left[\hat{\Sigma}^{\alpha}(\mu - \hat{\mu})\right]'\right\| \;&=\; \left\|\left[\left(\frac{\hat{\Sigma}}{N}\right)^{\alpha}\frac{(\mu - \hat{\mu})}{N}\right]'\right\| \\[6pt]
&\leq\; \left\|\left(\frac{\hat{\Sigma}}{N}\right)^{\alpha}\right\|\left\|\frac{(\mu - \hat{\mu})}{N}\right\| \\[6pt]
&=\; O_p\left(\frac{1}{\alpha\sqrt{TN}}\right)
\end{aligned}
$$

$$
\begin{aligned}
\left[\hat{\Sigma}^{\alpha}(\mu - \hat{\mu})\right]'\Sigma\left[\hat{\Sigma}^{\alpha}(\mu - \hat{\mu})\right] \;&=\; O_p\left(\left\|\left[\hat{\Sigma}^{\alpha}(\mu - \hat{\mu})\right]'\right\|^2\|\Sigma\|\right) \\[6pt]
&=\; O_p\left(\frac{\|\Sigma\|}{\alpha^2 TN}\right)
\end{aligned}
$$

Using the fact that $\|\Sigma\| = O(N)$ by Assumption A, we obtain that

$$
\left[\hat{\Sigma}^{\alpha}(\mu - \hat{\mu})\right]'\Sigma\left[\hat{\Sigma}^{\alpha}(\mu - \hat{\mu})\right] = O_p\left(\frac{N}{\alpha^2 TN}\right) = O_p\left(\frac{1}{\alpha^2 T}\right)
$$

$$
\begin{aligned}
\hat{x} - x \;&=\; \hat{\Sigma}^{\alpha}\hat{\mu} - \Sigma^{-1}\mu \\
\hat{\mu} \;&=\; (\hat{\mu} - \mu) + \mu \Rightarrow \\
\hat{x} - x \;&=\; \hat{\Sigma}^{\alpha}(\hat{\mu} - \mu) + \left(\hat{\Sigma}^{\alpha} - \Sigma^{-1}\right)\mu \Rightarrow
\end{aligned}
$$

$$
\left[\hat{\Sigma}^{\alpha}(\mu - \hat{\mu})\right]'\Sigma(\hat{x} - x) = \left[\hat{\Sigma}^{\alpha}(\mu - \hat{\mu})\right]'\Sigma\left[\hat{\Sigma}^{\alpha}(\mu - \hat{\mu})\right] + \left[\hat{\Sigma}^{\alpha}(\mu - \hat{\mu})\right]'\Sigma\left(\hat{\Sigma}^{\alpha} - \Sigma^{-1}\right)\mu
$$

$$
\begin{aligned}
\left(\hat{\Sigma}^{\alpha} - \Sigma^{-1}\right)\mu \;&=\; \left(\frac{\hat{\Sigma}}{N}\right)^{\alpha}\left[\frac{\Sigma}{N} - \frac{\hat{\Sigma}}{N}\right]\left(\frac{\Sigma}{N}\right)^{-1}\frac{\mu}{N} \\[6pt]
&=\; O_p\left(\frac{1}{\alpha\sqrt{TN}}\right)
\end{aligned}
$$

which implies that

$$
\begin{aligned}
(A) \;&=\; (\hat{x}-x)'\Sigma(\hat{x}-x) + O_p\!\left(\frac{2}{\alpha^2 T}\right) \\
&=\; (\hat{x}-x)'\Sigma(\hat{x}-x) + O_p\!\left(\frac{1}{\alpha^2 T}\right)
\end{aligned}
$$

Therefore, we obtain that

$$
\begin{aligned}
&E\!\left\{ \mu'\left[\left(\hat{\Sigma}^{\alpha}-\Sigma^{-1}\right)'\Sigma\left(\hat{\Sigma}^{\alpha}-\Sigma^{-1}\right)\right]\mu \right\} \\
&\sim\; E\!\left\{ (\hat{x}-x)'\Sigma(\hat{x}-x) \right\}
\end{aligned}
$$

if $\frac{1}{\alpha^2 T} \to 0$.

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
