# Peer review of "Regularized Maximum Diversification Investment Strategy†"

_econometrics, doi:10.3390/econometrics9010001_

Round 1
Reviewer 1 Report
Still, the biggest issue lingers on the contribution of this paper to the literature. In particular, as noted in the first-round review, Proposition 2 that this data-driven method depends on is directly from the existing literature. Research design, especially the empirical part, is also problematic. Empirical results should compare across different tuning parameter estimation methods in the literature, rather than using only the same tuning parameter method across different portfolios, as this paper mainly promotes a "new" tuning parameter calculation method. Such design of comparison speaks no valid information about the superiority of the "new" method, thus cannot demonstrate the contribution to the literature.
Besides, the exact equation for forecasting experiment is missing. The author answers that there are equations for simulation in the paper. These are different things. The explicit expression of the experiment setup is basic for empirical study.
Several questions on the first-round feedback are ignored by the author and not addressed. These questions focus on the fundamental aspects of this paper, which needs thorough attention.
Author Response
Good morning Dear Reviewer,
Thank you very much for your interest in my paper and for all the useful comments you have done.
I have seen your additional comments. Thank you for these additional comments.
I explain in the following cover letter how I address them.
Best regards,

Reviewer 2 Report
Dear author,
I have reviewed the revised manuscript and attest that all questions I had raised in my referee report have been properly answer to my full satisfaction.
As such, I recommend to accept this paper for publication.
Author Response
Good morning Dear Reviewer,
Thank you very much for your interest in my paper.
Best regards,
This manuscript is a resubmission of an earlier submission. The following is a list of the peer review reports and author responses from that submission.
Round 1
Reviewer 1 Report
The paper „Regularized Maximum Diversification Investment Strategy” investigates the use of regularization techniques to overcome numerical issues and to improve the performance of the so-called maximum diversification portfolio strategy. The author motivates the topic by the empirically observed mediocre out-of-sample performance of non-regularized optimized portfolios.
The paper is well-written and structured, the research design is appropriate and the analysis is thoroughly executed.
Regarding the level of novelty, the paper applies well-known techniques of regularizing the inverse covariance matrix to a new portfolio problem, namely the maximum diversification approach. Hence the paper is rather a combination of an existing portfolio allocation with well-studied statistical techniques to ameliorate estimation errors. The authors correctly cites a related paper that applies the same approach to the more classical mean-variance portfolio problem. While this hints at a medium level of novelty and innovation, the problem formulation and the suggested solutions are nevertheless highly relevant and interesting, both to practitioners and to financial economists. Moreover, the paper provides derivations of asymptotic properties of the proposed portfolio which, to the best of my knowledge, are novel and original. In the empirical section, the authors gives an insightful simulation study, as well as actual empirical results using real market data.
There are several things that I would like to point out and hope that the authors can improve the paper by following these guidelines and answering these questions:
- The study with real-world data only uses Fama-French portfolio, likely because they are easily available. Since portfolio optimization is done on single stocks, not on portfolios, I would have preferred to see a single-stock example, e.g. using different subset of the S&P500 constituents to check the impact of the sample size also with real data.
- Many papers in the area of regularization for portfolio optimization use monthly data and up to 100 assets (dimensions). For an application where the ration of dimensions (N) to time points for estimation (T) is highly relevant, I find it unfortunate that the examples are constructed in a way that N is similar to T, solely because T is low when using monthly data. In reality, one would use daily returns, and maybe 3-5 years of data. Then T>750 and N is much smaller in most examples. While I see that using Fama-French industry portfolios with montly returns is tempting, it lacks of relevance for real world applications. At least in the simulation parts, one could try the case where T=1000 and N is between 100 and up to 1000.
- The author mentions that the method is motivated by the desire to avoid short-selling constraints. However, if short-selling would be prohibited, the problem would be already regularized by the long-only constraint, and it would be interesting to see whether the proposed method still improves the portfolio performance. The same holds true, again, for most papers in the literature on covariance shrinkage for portfolio optimization, e.g., the works of Ledoit and Wolf. While it is obvious that using a long-only constraint, or even something more realistic like all weights being between 0 and 10% of the total wealth, will diminish the impact of using regularization, and honest analysis that investigates whether in this case the use of regularization still improves things and can be recommended to the practitioner would be highly appreciated.
- The abstract could be improved by adding one sentence on what the max. diversification strategy is and why it is interesting.
- In the empirical part, I could not find information about how many data points are used for estimation and how many rolling windows are used. This is very relevant, in order to judge the concentration ratio T/N.
- The portfolio evolution in the out-of-sample exercise should be plotted over time, for the different regularization methods. Also, other performance metrics than Sharpe ratio, returns and vola should be provided, in particular the maximum drawdown and also the turnover generated by these strategies.
- A vast improvement of using regularization or shrinkage in practice comes from the fact that the covariance matrices are also more stable across time, leading to less portfolio turnover. The authors would profit from providing also the performance net of returns, using proportional transaction costs with a realistic level of 10-20 basis points.
- The author implicitly claims that the portfolio with the highest Sharpe ratio is the most diversified one. This is certainly not true: if there is a single asset with a vastly higher Sharpe ratio than any other method, all portfolios using several assets will have a lower Sharpe ratio although they are more diversified, in terms of not being concentrated in just one asset.
- While doing a FF-regression is standard in many finance papers, I do not really see the value here of doing so here. Does it even make economically sense to claim that using different covariance estimators will create exposure to different Fama-French factors? I do not think this relationship is stable across time or statistically significant in many periods.
- The paper should be improved by a checking the grammar and some typos again more carefully. For example, the second paragraph in Chapter 1 seems to be incomplete (after the long parenthesis the main sentence does not continue). Or “the covariance matrix […] need to be estimated” should be “needs”. Also the frequent use of the term “So, …” to start a sentence should be avoided because it is too colloquial (see also in the abstract). The first few sentences in Chapter 2 also need more attention. For example, there is the sentence “Rf empirically with monthly data to be the mean […]” – I have problems deciphering what this is supposed to mean.
- The second half of p.13 seems to be a mere repetition of earlier parts, maybe it was a copy-and-paste leftover.
Overall I think the paper has academic merit and the topic is of great interest to academics and practitioners in the field of portfolio optimization. I am convinced that the author could improve the paper by addressing the questions and problems raised above.
Therefore I recommend to invite the author for resubmission after a minor revision.
Reviewer 2 Report
Please see details in the attached report.

Reviewer 3 Report
Please see attached report.
